# Acute Traumatic Endotheliopathy in Isolated Severe Brain Injury and Its Impact on Clinical Outcome

**DOI:** 10.3390/medsci6010005

**Published:** 2018-01-16

**Authors:** Venencia Albert, Arulselvi Subramanian, Deepak Agrawal, Hara Prasad Pati, Siddhartha Datta Gupta, Asok Kumar Mukhopadhyay

**Affiliations:** 1Department of Laboratory Medicine, Jai Prakash Narayan Apex Trauma Centre, All India Institute of Medical Sciences (AIIMS), New Delhi 110029, India; albertvenencia@gmail.com (V.A.); mukhoak1953@gmail.com (A.K.M.); 2Department of Neurosurgery, Jai Prakash Narayan Apex Trauma Centre, All India Institute of Medical Sciences (AIIMS), New Delhi 110029, India; drdeepak@gmail.com; 3Department of Hematology, Jai Prakash Narayan Apex Trauma Centre, All India Institute of Medical Sciences (AIIMS), New Delhi 110029, India; harappati@yahoo.co.in; 4Department of Pathology, Jai Prakash Narayan Apex Trauma Centre, All India Institute of Medical Sciences (AIIMS), New Delhi 110029, India; sduttagupta@gmail.com

**Keywords:** traumatic, brain injuries, endothelium, impact

## Abstract

Study design: Prospective observational cohort. Objective: To investigate the difference in plasma levels of syndecan-1 (due to glycocalyx degradation) and soluble thrombomodulin (due to endothelial damage) in isolated severe traumatic brain injury (TBI) patients with/without early coagulopathy. A secondary objective was to compare the effects of the degree of TBI endotheliopathy on hospital mortality among patients with TBI-associated coagulopathy (TBI-AC). Methods: Data was prospectively collected on isolated severe TBI (sTBI) patients with Glasgow Coma Scale (GCS) ≤8 less than 12 h after injury admitted to a level I trauma centre. Isolated sTBI patients with samples withdrawn prior to blood transfusion were stratified by conventional coagulation tests as coagulopathic (prothrombin time (PT) ≥ 16.7 s, international normalized ratio (INR) ≥ 1.27, and activated partial thromboplastin time (aPTT) ≥ 28.8 s) and non-coagulopathic. Twenty healthy controls were also included. Plasma levels of thrombomodulin and syndecan-1 were estimated by ELISA. With receiver operating characteristic curve (ROC) analysis, we defined endotheliopathy as a syndecan-1 cut-off level that maximized the sum of sensitivity and specificity for predicting TBI-AC. Results: Inclusion criteria were met in 120 cases, with subjects aged 35.5 ± 12.6 years (88.3% males). TBI-AC was identified in 50 (41.6%) patients, independent of age, gender, and GCS, but there was an association with acidosis (60%; *p* = 0.01). Following isolated sTBI, we found insignificant changes in soluble thrombomodulin (sTM) levels between patients with isolated TBI and controls, and sTM levels were lower in coagulopathic compared to non-coagulopathic patients. Elevations in plasma syndecan-1 (ng/mL) levels were seen compared to control (31.1(21.5–30.6) vs. 24.8(18.5–30.6); *p* = 0.08). Syndecan-1(ng/mL) levels were significantly elevated in coagulopathic compared to non-coagulopathic patients (33.7(21.6–109.5) vs. 29.9(19.239.5); *p* = 0.03). Using ROC analysis (area under the curve = 0.61; 95% Confidence Interval (CI) 0.50 to 0.72), we established a plasma syndecan-1 level cutoff of ≥30.5 ng/mL (sensitivity % = 55.3, specificity % = 52.3), with a significant association with TBI-associated coagulopathy. Conclusion: Subsequent to brain injury, elevated syndecan-1 shedding and endotheliopathy may be associated with early coagulation abnormalities. A syndecan-1 level ≥30.5 ng/mL identified patients with TBI-AC, and may be of importance in guiding management and clinical decision-making.

## 1. Introduction 

### 1.1. Background 

About 25% of traumatic brain injury (TBI) cases are complicated by haemorrhagic shock, which increases mortality by up to three-fold [1]. TBI-associated coagulopathy (TBI-AC) is linked with progression of haemorrhagic lesions [2], with an overall prevalence of 32.7% [3], despite the lack of extracranial bleeding and restricted fluid infusion in patients with isolated TBI as compared to patients with multisystem trauma [4].

Acute trauma-induced coagulopathy (ATIC) is strongly associated with poor outcome, increased transfusion requirements, organ dysfunction, and Intensive care unit (ICU) stay [3,5,6], presenting in both hypocoagulable and hypercoagulable states [7]. Pathophysiological mechanisms of ATIC are indistinct, due to its complex, multifactorial pathophysiology, the shortcomings of routine diagnostic tests [8,9], and inconsistent clinical terminology [10]. Additionally, it is also debated whether the TBI-AC differs from multisystem trauma [3,11,12,13]. 

Brohi et al. suggested that local endothelial damage, as suggested by high soluble thrombomodulin (sTM) levels (a marker of endothelial damage), in the early stage of trauma causes activation of coagulation factors, and boosts thrombin generation [14,15]. 

A review of studies on multisystem trauma suggests that neurohumoral activation propels coagulopathy through endothelial destruction, particularly by glycocalyx disintegration [16,17,18]. Vasoactive catecholamines, inflammatory mediators, and the injury itself results in redistribution of blood flow, activation and mobilization of platelets, and upregulation of endothelial procoagulant/profibrinolytic factors, leading to a localized procoagulant milieu [19,20,21,22]. 

Endothelial glycocalyx (EGL), which was first postulated by Danielli in 1940 [23], is a fragile, 0.2-μm- to 1-µm-thick, negatively charged antiadhesive and anticoagulant carbohydrate-rich surface layer [24]. It projects into the lumen of the blood vessel from the endothelial surface [25]. EGL is hypothesized to be damaged before the endothelium and therefore serves as sensitive indicator of injury [26]. EGL shedding can be quantified by a concurrent increase in circulating levels of EGL components, particularly syndecan-1 [16,27,28,29].

Over the past decades, attempts have been made to utilize endothelial glycocalyx in vascular physiology and pathology through the measurement of each element of glycocalyx circulating in the blood. Post-trauma elevations in syndecan-1 levels have been associated with coagulopathy and mortality [22,28,30], suggesting that degradation of the endothelial glycocalyx may contribute to the development of ATIC [16,22,31,32].

We therefore hypothesized that severe TBI would disrupt endothelial integrity and glycocalyx degradation by promoting upregulation of syndecan-1 and thrombomodulin and exacerbation of coagulation abnormalities, predisposing the patients to poor outcome. 

### 1.2. Objectives 

The primary objective of this study was to study the difference in syndecan-1 and soluble thrombomodulin (sTM) levels in severe isolated TBI patients with/without early coagulopathy. A secondary objective was to compare the effect of degree of endotheliopathy on the mortality rate among patients with/without TBI-AC. 

## 2. Materials and Methods 

### 2.1. Study Design, Participants and Setting 

This prospective observational diagnostic cohort study was performed with the approval from the institutional ethics committee (Ref. IESC/T-431/30.11.2012, OT-1/27.01.2016). Written informed consent was taken prior to sampling from the patients or their legal authorized guardians for all individual participants included in the study. 

Trauma patients who presented at the Emergency Department (June 2014 to December 2016) were assessed by the attending neurosurgeon for severity of head injury using the Glasgow Coma Scale (GCS), Abbreviated Injury Scale (AIS), and Injury Severity Score (ISS) adjunct to computer-aided tomography (CT) performed as per the routine standard assessment. Consent was obtained from patient or a legally authorized representative. We screened 514 severe TBI patients (aged 20–50 years) and 120 isolated sTBI patients were recruited. Exclusion criteria of the study were: associated injury, transfusion (fluid/blood products) prior to sample collection, pregnancy, co-morbidities (liver disease, diabetes mellitus), history of (h/o) haemorrhagic disorder, preinjury anticoagulant medication, on-admission hypotension (systolic arterial blood pressure <80 mm/Hg), secondary admissions, and clinical evidence of brain death (GCS = 3) (Figure 1). Patient management was carried out according to protocols used in our institute 

### 2.2. Defining Isolated TBI

Head injuries [33] were defined as severe TBI if they met any of the any of the three criteria: (1) Glasgow Coma Scale (GCS) ≤8; (2) head abbreviated injury scale (AIS) ≥2; and (3) intracranial haematoma on CT scan of the head (cerebral contusion; subarachnoid, subdural, or epidural haemorrhages). Severe TBI without any extracranial injuries was defined as isolated severe traumatic brain injury (sTBI).

### 2.3. Defining TBI Associated Coagulopathy

Patients were stratified by conventional coagulation tests as coagulopathic (International normalized ratio (INR) ≥1.27 and/or Prothrombin time (PT) ≥16.7 and/or activated partial thromboplastin time (aPTT) ≥ 28.8) on hospital admission. This definition of ATIC was chosen based on the study by Greuters et al. [34] and our institutional current clinical practice and experience.

### 2.4. Sample Size

Sample size was calculated with tissue injury as the driver for early trauma-induced coagulopathy. Assuming mean ± standard deviation of sTM [22] in coagulopathy patients as 2.5 ± 1.5 and in patients who do not develop coagulopathy as 1.5 ± 1.0, to detect this difference with 90% power and 95% confidence limits and anticipated TBI-AC rate being >40%, we enrolled 120 TBI patients.

A control group of 20 age/gender-matched healthy controls (HCs) was included to assess the baseline of biomarkers of tissue/endothelial cell damage and glycocalyx degradation. 

### 2.5. Sample Processing and Analysis

To minimize the impact of confounding therapy, a one-time sample was collected in the emergency department (≤12 h of injury) 

#### 2.5.1. Blood

Three millilitres of intravenous blood samples were collected in an ethylenediaminetetraacetic acid (EDTA) vial (1:4) as part of routine blood analysis. Plasma was separated by centrifugation and stored at −20 °C until analysis 

#### 2.5.2. Enzyme Linked Immunosorbent assay 

Soluble biomarkers of tissue/ endothelial cell (Human Thrombomodulin ELISA Kit (CD141), catalogue number: ab46508; Abcam plc., San Francisco, CA, USA) and glycocalyx damage (Human Syndecan-1 ELISA Kit (CD138); cat: ab46506; Abcam plc., San Francisco, CA, USA), in duplicate by commercial enzyme-linked immunosorbent assays (ELISAs) were used according to the manufacturer’s recommendations. 

### 2.6. Study Variables 

All data pertaining to patient demographics (age, gender and mode of injury), clinical course (time of injury and time of hospital arrival, vital signs on admission, medical/medication history and CT findings), routine laboratory investigations, blood product administration, and surgical interventions was monitored and recorded. Acidosis was defined as pH < 7.3 and HCO_3_ < 20 mEq/L. Hypoperfusion was defined as arterial base deficit >6 mmol/L. The primary patient outcome measure was incidence of post-trauma mortality (at 48 h and 30 days) that occurred in the hospital. Duration of ICU and hospital stay, need for transfusion of blood and blood products, and sepsis (blood culture) were secondary outcome measures. 

### 2.7. Statistical Methods

Statistical analysis was performed for the comparison of parameters for two outcomes: firstly, TBI-associated coagulopathy and no TBI-associated coagulopathy; and secondly, survivors and non-survivors during the hospital stay. Summary statistics were used to describe continuous variables as mean ± standard deviation or median [interquartile range (IQR)] and categorical data were presented as frequency (%). Analysis was performed using STATA 11.0 statistical software (StataCorp LLC, College Station, TX, USA). Univariate analysis of the continuous variables between the coagulopathic and non-coagulopathic study group was assessed using *t*-tests/Wilcoxon rank-sum. Chi χ^2^ tests or Fisher’s exact test was used to compare categorical variables. Statistical significance was set at the *p* < 0.05 level. The correlation between continuous variables was explored with Spearman’s rank correlation coefficient. Receiver operating characteristic (ROC) curve analysis was used to generate optimal cut-offs for endothelial damage markers found to be significant in univariate analysis for identification of isolated sTBI patients with early coagulopathy.

## 3. Results 

One hundred twenty subjects, with an average age of 35.5 ± 12.6 years, including 106 (88.3%) males, were included in the study. Road traffic accident was the leading cause of injury (overall 83 (69.2%)). Average time taken from injury to admission was two hours (range 1–5). Based on the TBI-AC definition, patients were classified into two subgroups: coagulopathic patients (50, 41.6%) and non-coagulopathic patients (70, 58.4%).

Although most baseline pre-injury variables were identical between the two groups, coagulopathic isolated sTBI patients demonstrated a more severe degree of hypoperfusion, anaemia, and thrombocytopenia compared to non-coagulopathic patients; however this variation was statistically insignificant (Table 1). 

Overall, 60.3% patients in the coagulopathic isolated sTBI patients had acidosis, which was (significantly) two-fold higher than in the non-coagulopathic group (30%) (*p* = 0.01). 

On admission 16 (13.3%) had anaemia, and 15 (12.5%) had thrombocytopenia, following isolated sTBI. Full blood count parameters were not significantly associated with the development of early coagulopathy, however a higher frequency of patients had on admission anaemia and thrombocytopenia in the coagulopathy group, as compared to those who did not develop coagulopathy (Table 1). 

### 3.1. Acute Coagulopathy, Glycocalyx Shedding, and Endothelial Disruption of Traumatic Brain Injury 

sTM levels did not vary in response to brain injury, however in coagulopathic patients, an early albeit insignificant decline was observed in comparison to non-coagulopathic isolated sTBI patients and the control group. Elevated shedding of syndecan-1 in isolated sTBI patients compared to HCs (31.3(21.3–54.3) ng/mL vs. 24.8(21.5–30.6) ng/mL; *p* = 0.08) was observed. 

Also, significantly elevated plasma syndecan-1 levels were seen in patients who developed early coagulopathy (33.7(21.6–109.5) ng/mL) as compared to those who did not (29.9(19.2–39.5) ng/mL; *p* = 0.03), and the control group (*p* = 0.04) (Figure 2 and Table 2). 

ROC curve analysis revealed a cut-off level of syndecan-1 with a sensitivity of 55.3% and specificity of 52.2%. We determined our cut-off level of ≥30.5 ng/mL as the level of syndecan-1 for prediction of early coagulopathy as the outcome (area under the curve = 0.62; 95% Confidence interval (CI) 0.50 to 0.72) (Figure 3).

Based on the established cut-off, isolated sTBI patients were stratified as having TBI endotheliopathy (51.7%; syndecan-1 ≥ 30.5 ng/mL) or no TBI endotheliopathy (48.3%; syndecan-1 < 30.5 ng/mL). The groups did not vary in age, gender frequency, or GCS. Mode of injury did differ significantly; patients who succumbed to their injuries due to a fall (75% vs. 25%) and assault (100%) had a higher frequency of endotheliopathy (*p* < 0.03). Complete blood count and coagulation assays fell in normal ranges; however, mean haemoglobin levels (12.2 ± 2.60 vs. 13.1 ± 2.52; *p* = 0.06) and red blood cell counts (4.1 ± 0.85 vs. 4.5 ± 0.78; *p* = 0.05) were significantly lower in patients with TBI endotheliopathy as compared to those without.

Soluble TM levels in patients TBI endotheliopathy were nearly two times higher (14.7(6.8–86.7) ng/mL) than in those without endotheliopathy (7.2(4.2–85.9) ng/mL) (*p* < 0.20). Significantly, sTM levels in coagulopathic patients with endotheliopathy were nearly four times higher (median 20.2 ng/mL) than in those without endotheliopathy (5.5 ng/mL) (*p* < 0.002). Table 3. Overall, there was a weak positive correlation between syndecan-1 and sTM levels (Spearman’s *ρ* correlation coefficient = 0.23; *p* = 0.01). In patients with TBI-AC, a positive moderate correlation between syndecan-1 and sTM levels (Spearman’s *ρ* = 0.50; *p =* 0.0007) was seen. 

### 3.2. Clinical Outcome 

Overall mortality was 30% (36 out of 120). The mortality rate in patients who developed TBI-AC was two times higher in comparison to those who did not (22(44) vs. 14(20); *p* = 0.005). Although elevated median (IQR) hospital and ICU length stay was associated with early coagulopathy, this was not statistically significant. TBI-AC also results in a significant increase in transfusion requirements (packed red blood cells (PRBC) (*p* = 0.002); platelet rich plasma (PRP) (*p* = 0.001); fresh frozen plasma (FFP) (*p* = 0.0007)) (Table 1). 

Multivariable logistic regression analysis with seven covariates (age, GCS, mechanical ventilation, shock index, sepsis, infection, and organ failure) was performed. Controlling for these variables, we found that TBI-AC was an independent predictor with an increased likelihood of in hospital mortality (adjusted odds ratio = 4.73; 95% CI 1.68 to 13.3). 

Overall, in isolated sTBI patients the median sTM levels were slightly elevated in non-survivors as compared to survivors, but this variation was statistically insignificant (13.2(4.0–86.9) ng/mL vs. 10.1(5.12–83.8) ng/mL *p* = 0.69). Syndecan-1 levels however were significantly elevated in non-survivors as compared to survivors (37.5(30.4–138.9) ng/mL vs. 23.5(19.2–40.1) ng/mL *p* < 0.0001).

On comparing the effect of TBI endotheliopathy (based on our empirically-derived cut off) on clinical outcomes in isolated sTBI patients, we observed no variation in the length of stay between those with and without endotheliopathy. A significant two-fold increase in transfusion requirements was seen in patients with TBI endotheliopathy. Similarly, mortality rate at 47 h after injury was almost three times higher than in endotheliopathy vs. no endotheliopathy (20.3% vs. 6.6% *p* = 0.03) patients. Mortality rate at 30 days after injury was also twice as high in patients with TBI endotheliopathy (4.1%) as compared to patients without endotheliopathy (16.4%) (Table 4). 

## 4. Discussion 

This study characterized the blood biomarkers of endothelial damage and glycocalyx disruption following isolated severe TBI to identify a potential mechanistic link between endotheliopathy and coagulopathy. In our study, early coagulopathy after isolated TBI was associated with significant elevations in circulating biomarkers of glycocalyx degradation, whilst the marker for endothelial damage did not vary significantly. 

A component of the glycocalyx, syndecan-1, is released from the tissue and can be detected in the circulating blood of isolated TBI patients, directly implicating the endothelial surface layer as a potential pathophysiological factor in the development of acute TBI-AC. 

We found that sTM levels did not vary between isolated TBI patients and HCs in the cohort as a whole.

Early changes in endothelial disruption and damage markers were observed in coagulopathic patients compared to non-coagulopathic isolated sTBI patients. Interestingly, sTM levels declined in coagulopathic vs. non-coagulopathic patients, whereas syndecan levels were elevated in coagulopathic patients as compared to non-coagulopathic patients. This could be explained by fact that levels of sTM reflect a deeper component of the endothelium, whereas syndecan-1 reflects superficial endothelial damage. sTM is only released upon direct endothelial disruption, which cannot explain its early decline in the circulating sTM levels, as secondary endothelial cell damage has not yet occurred [26,35], and circulatory sTM is being consumed in the protein C (PC)–thrombomodulin pathway.

The average time taken from injury to sample collection in our study was two hours (IQR 1–5). Previously significantly elevated glycocalyx shedding (851.0 pg/mL vs. 715.5 pg/mL, *p =* 0.03) has been reported to occur 15 min after injury in a porcine model. Syndecan-1 elevation was also associated with elevated von willebrand factor (vWF, 784 U/mL vs. 645 U/mL, *p* = 0.01) and soluble vascular cell adhesion molecule 1 (sVCAM-1, 9.47 ng/mL vs. 5.90 ng/mL, *p* =0.02) [36]. 

Soluble TM level in coagulopathic patients with endotheliopathy was nearly four times higher (median 20.2(8.0–88.0) ng/mL) than in patients without endotheliopathy (5.5(1.6–10.6) ng/mL) (*p* < 0.002). Similarly, Rodriguez et al. [35] reported 1.5 times higher sTM levels in patients with endotheliopathy (syndecan-1 level ≥ 40 ng/mL) than in those without endotheliopathy (syndecan-1 level < 40 ng/mL) (*p* < 0.001). They also reported a positive moderate correlation between syndecan-1 and sTM levels (Spearman’s r correlation coefficient = 0.50; 95% CI 0.42 to 0.56). However, we observed an overall positive weak correlation between syndecan-1 and sTM levels, but a positive moderate correlation (Spearman’s *ρ* = 0.50; *p* = 0.0007) in patients with TBI-AC. 

Syndecan-1 levels have been studied previously in major trauma [28,29,30] and/or haemorrhagic shock [37], and sepsis [17,18,26,38]. Elevated syndecan-1 levels [39] are strongly correlated with inflammation, hypocoagulability, and poor outcome [22]; and resuscitation with plasma results in a fall in syndecan-1 levels [30]. Collectively, the above studies indicate that endothelial damage significantly contributes to patient outcome [40]. 

Johansson et al. [22] reported that elevation of syndecan-1 level is a marker of high endothelial glycocalyx degradation in severe injury, which is associated with high sympathoadrenal activity and poor outcome. Increasing injury severity in patients with elevated syndecan-1 levels was associated with gradual PC depletion, increasing sTM, and hyperfibrinolysis, suggesting that EGL degradation may be concomitant to ATIC. 

The results of a study by Battista et al. [41] were consistent with others who correlated TBI with an immediate, procoagulant, and hyperfibrinolytic state. They found that on admission procoagulant hyperfibrinolysis markers strongly contributed to unfavourable outcome, whereas the anticoagulant indices sTM were insignificant.

Shedding of the glycocalyx into the circulation is reported to indirectly contribute to a hypercoagulant state by increasing circulating concentrations of damage-associated molecular patterns (DAMPs) [42] or may lead to a hypocoagulant state through endogenous heparinization [16]. Chapman et al. [37] reported that an endothelial cell-centred inflammation–coagulation pathway induces coagulation imbalance, which leads to ATIC by global coagulation activation followed by enhanced anticoagulation and fibrinolysis. A prospective study conducted by Ostrowski and Johansson [16] described endogenous heparinization in 5.2% patients post trauma, as identified by thromboelastography (TEG). A degree of endogenous heparinization also associated with plasma syndecan-1 levels, linking endothelial glycocalyx degradation with autoheparinization and suggesting that glycocalyx induces autoheparinization, causing ATIC. Endogenous heparinization correlated with four-fold higher circulating levels of syndecan-1 (116(78–140) ng/mL vs. 31(18–49) ng/mL, *p* = 0.020) and increased sTM (4.1(3.7–4.4) ng/mL vs. 1.7(1.0–3.3) ng/mL; *p* = 0.028). However, the proportion of patients with TBI among patients with or without endogenous heparinization was comparable in their study. 

Here, we report a significant association of elevated plasma syndecan-1 levels with TBI-AC (33.7(21.6–109.5) ng/mL vs. 29.9(19.2–39.5) ng/mL; *p* = 0.04), supporting the notion that glycocalyx shedding contributes mechanistically to TBI-AC. 

In another study of 404 trauma patients, Ostrowski et al. [43], reported that patients with coagulopathy had higher plasma syndecan-1 levels (26 (14–79) ng/mL vs. 23 (12–50) ng/mL, *p =* 0.01). Endothelial cell junction and glycocalyx damage were independently associated with hypocoagulability by rapid thromboelastography (rTEG). Plasma sTM levels however declined in coagulopathic patients (5.0(3.7–7.3) vs. 5.3(3.9–7.0); *p* = 0.66), similar to our study. 

Iseki et al. [44] examined the expression of syndecan core proteins genes in a mouse model and found the upregulation of syndecan messenger RNA (mRNA) to be closely related to the reactivity of astrocytes, as they are the primary source of syndecan. Expression levels of syndecan peaked at day 7 after injury in their study. 

Yokota et al. [38] demonstrated that cerebral tissue injury is often accompanied by cerebral endothelial activation. In their study the serum level of sTM in patients with delayed traumatic intracerebral haematoma (DTICH) was significantly higher than in patients without DTICH. The serum level of sTM in focal brain injury (3.84 ± 1.54 to 4.12 ± 1.46 U/mL) was higher than that in diffuse axonal injury (ranging from 2.96 ± 0.63 to 3.67 ± 1.70 U/mL), indicating that serum sTM is a good indicator of the cerebral endothelial injury and of endothelial activation in severe head injury. 

Endotheliopathy of trauma is a term proposed by Holcomb and Pati [45] in an attempt to characterize a syndrome that arises after injury, probably prompted by EGL breakdown. Rodriguez et al. [35] in their study defined endotheliopathy of trauma based on a syndecan-1 level ≥40 ng/mL with maximum sensitivity and specificity in predicting 24 h in-hospital mortality in severely injured patients. Based on our findings we derived a clinical cut-off for a plasma syndecan-1 level of ≥30.50 ng/mL. Interestingly, the cutoff for the highest syndecan-1 quartile in the Johansen [26] study was 240 ng/mL, which was identical to the cutoff utilized by Puskarich et al. [46] in their cohort of patients with severe sepsis to predict the primary outcome of intubation. Recently, Naumann et al. [47] defined endotheliopathy as values above the 97.5th percentile of HCs for either syndecan-1, thrombomodulin, or both, in trauma patients (ISS ≥ 8). Based on their statistical modelling, glycocalyxshedding and endothelial cell injury were estimated to occur at approximately 5 and 8 min following injury, respectively. However, this estimation was made based on several assumptions that have the potential to be biased.

We observed that TBI endotheliopathy was significantly associated with an increased need for transfusion and risk of early mortality. Similarly, in the study by Rodriguez et al. [35] most patients (71.7%) in the endotheliopathy group needed blood transfusions and required four times more units of blood products in the initial 24 h than patients without endotheliopathy. Johansen et al. [26], directly compared, syndecan-1 and sTM in a cohort of critically ill patients. Although both markers were significant predictors of mortality in their univariate analysis, only sTM remained a significant predictor in the adjusted model. Puskarich et al. [46] found that patients with high levels of Syndecan-1 tended to have a lower platelet count, higher bilirubin, and a higher sequential organ failure assessment score (SOFA score). High levels were also associated with an increased risk of intubation, a longer length of stay and higher mortality. 

Battista et al. [41] reported a strong correlation of syndecan-1 with unfavourable outcome at both admission and at 24 h following TBI. 

Overall, our results are consistent with a relationship between poor outcome and biomarker indices of endotheliopathy over the first 24 h of hospitalization in isolated TBI patients. 

Our results may be limited, since we did not perform a follow up assessment of the endothelial damage markers to study the time course of endotheliopathy in isolated sTBI. This was due to the fact that we wanted to eliminate the confounding effect of various interventions given for patient management [48]. 

## 5. Conclusions

We identified a potential mechanistic link between endotheliopathy and coagulopathy. Early coagulopathy, with an incidence of 41.6%, was an independent predictor of mortality (odds ratio (OR) = 4.73; 95% CI 1.68–13.3). TBI-associated coagulopathy was associated with significant elevations in plasma syndecan-1, a surrogate marker of glycocalyx degradation (endotheliopathy of trauma), whilst thrombomodulin levels did not vary significantly. High syndecan-1 levels of ≥30.50 ng/mL were significantly associated with an increased need for transfusion and risk of early mortality. Therefore, we recommend measuring syndecan-1 to identify endotheliopathy-associated early coagulopathy following brain trauma. This identification will aid in implementing early clinical interventions aimed at protecting and repairing the endothelium to attenuate traumatic endotheliopathy and potentially improve outcomes in isolated sTBI patients. 

## Figures and Tables

**Figure 1 medsci-06-00005-f001:**
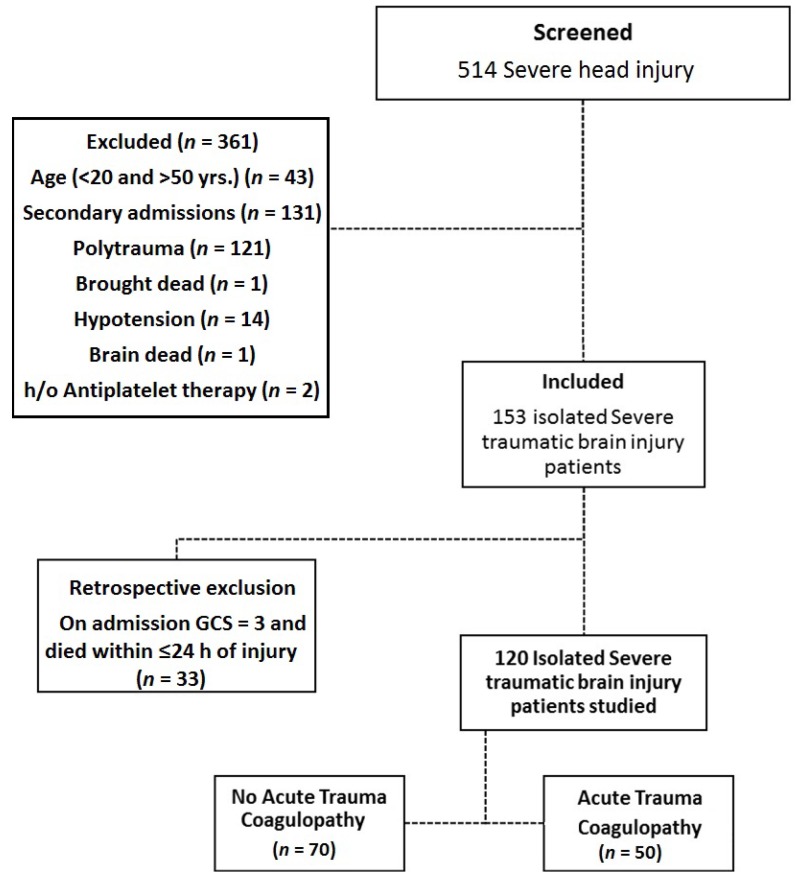
A Consolidated Standards of Reporting Trials (CONSORT)-like diagram showing the selection process used to identify traumatic brain injury patients for inclusion in the study. GSC: Glasgow Coma Scale; h/o: history of.

**Figure 2 medsci-06-00005-f002:**
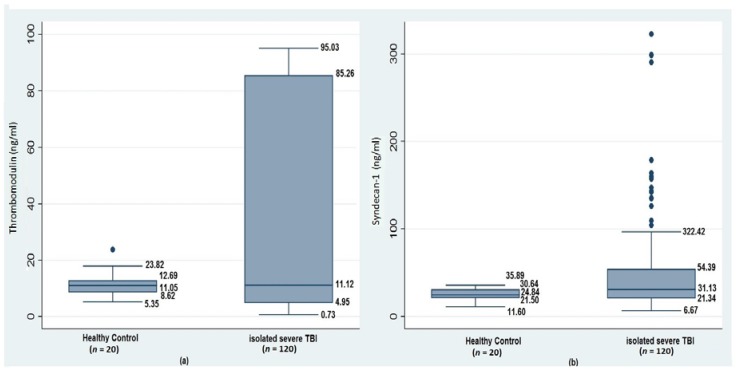
Comparison of plasma levels of (**a**) thrombomodulin ng/mL and (**b**) syndecan-1 ng/mL between healthy controls (HCs) and patients with isolated severe TBI (sTBI).

**Figure 3 medsci-06-00005-f003:**
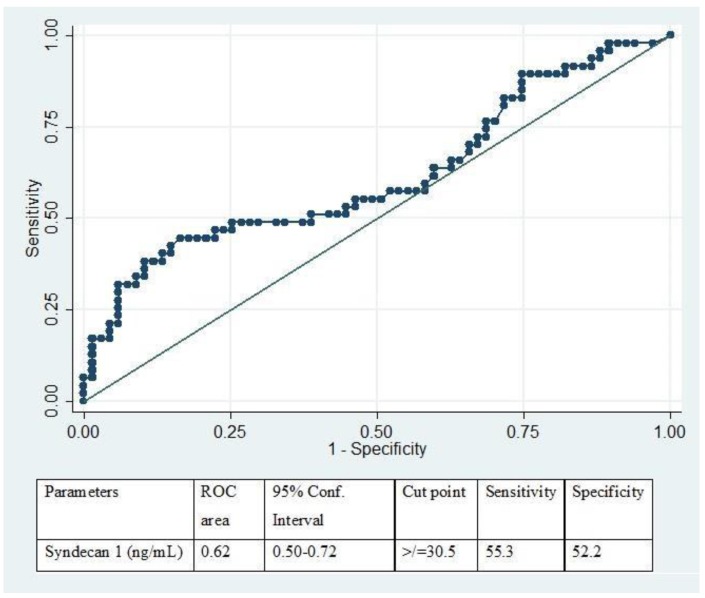
Receiver operating characteristic (ROC) curve analysis to establish a clinical cut-off of plasma syndecan-1 levels to identify TBI-associated coagulopathy.

**Table 1 medsci-06-00005-t001:** Demographic and clinical pattern of the study cohort overall and as stratified by the presence of traumatic brain injury-associated coagulopathy (TBI-AC).

Clinical Parameters		Overall (*n* = 120)	Study Group (*n* = 120)		No TBI-AC vs. TBI-AC *p*-Value
			No TBI-AC (*n* = 70)	TBI-AC (*n* = 50)	
Age * (years)		35.5 ± 12.6	36.0 ± 12.89	34.8 ± 13.35	0.62
Gender ^†^	Male	106(88.3)	61(87.1)	45(90.0)	0.63
	Female	14(11.6)	9(12.9)	5(10.0)	
Mode of injury ^†^	RTA	83(69.2)	48(68.5)	35(70.0)	0.83
	Fall	21(17.5)	12(17.1)	9(18.0)	
	Assault	2(1.67)	2(2.8)	0	
	Miscellaneous	14(11.7)	8(11.4)	6(12.0)	
Time taken from injury to admission * (h)		5(1–5)	2(1–5)	2(1–4)	0.71
Mechanical ventilation ^†^	No	44(36.7)	23(32.8)	21(42.0)	0.30
	Yes	76(63.3)	47(67.2)	29(58.0)	
Systolic BP^*^ (mm/Hg)		139.5 ± 26.4	143.1 ± 3.2	134.3 ± 3.4	0.07
Glasgow Coma Scale		7(5–8)	7(5–8)	6(5–7)	0.09
Hypoperfusion ^†^ (*n* = 72)	No	52(72.2)	33(78.5)	19(63.3)	0.15
	Yes	20(27.8)	9(21.6)	11(36.4)	
Acidosis ^†^ (*n* = 68)	No	39(57.4)	28(70.0)	11(39.2)	0.01
	Yes	29(42.6)	12(30.0)	17(60.3)	
Haemoglobin (g/dL)		12.5 ± 2.69	12.7 ± 2.50	12.3 ± 2.94	0.35
Anaemia ^†^ (Hb < 9gm/dL)	No	104(86.7)	61(87.1)	43(86.0)	0.85
	Yes	16(13.3)	9(12.9)	7(14.0)	
Red blood cell count (10^6^ cumm)		4.3 ± 0.85	4.4 ± 0.78	4.1 ± 0.93	0.06
Haematocrit (%)		39.7 ± 7.82	40.4 ± 7.13	38.6 ± 8.67	0.20
Platelet count * (10^3^/cumm)		193(134.5–250.5)	196.5(138.0–248.0)	184.5(116.0–251.0)	0.53
Thrombocytopenia ^†^ (plt < 100 × 10^3^ cumm)	No	105(87.5)	63(90)	42(84)	0.32
	Yes	15(12.5)	7(10)	8(16)	
White blood cell count * (4.0–11.0 × 10^3^/cumm)		14.3(10.8–18.7)	13.9(11.0–19.1)	14.7(10.8–18.4)	0.79
Prothrombin time (s)		16.0 ± 6.1	13.9 ± 1.1	18.8 ± 8.51	**<0.0001**
activated partial thromboplastin time (aPTT) (s)		29.6 ± 13.7	24.1 ± 2.4	37.3 ± 18.5	**<0.0001**
International normalized ratio (INR)		1.0 ± 0.36	1.0 ± 0.12	1.4 ± 0.46	**<0.0001**
Hospital length of stay ^†^ (days)		8(4–20)	7(2–16)	9(5–21)	0.73
ICU length of stay ^†^ (days)		5(3–9)	5(3–8)	5(2–9)	0.91
Transfusion requirements ^†^	PRBC (*n* = 57)	0(0–4)	0(0–3)	2(0–6)	1.0
	FFP (*n* = 34)	0(0–2.5)	0(0–0)	0(0–4)	
	PRP (*n* = 25)	0(0–0)	0(0–0)	0(0–4)	
Mortality ^†^	No	84(70)	56(80.0)	28(56.0)	**0.005**
	Yes	36(30)	14(20.0)	22(44.0)	

* Continuous variables are reported as means ± standard deviations or median (interquartile range); ^†^ Categorical variables are reported in terms of frequency (percentage). Bold font depicts significant *p*-value. RTA: road traffic accident; BP: blood pressure; Hb: haemoglobin; cumm: per cubic millimetre; plt: platelet count; ICU: intensive care unit; PRBC: packed red blood cells; FFP: fresh frozen plasma; PRP: platelet rich plasma.

**Table 2 medsci-06-00005-t002:** Correlation of glycocalyx shedding and endothelial disruption with acute TBI-associated coagulopathy.

Parameter	HC (*n* = 20)	Study Group (*n* = 120)		*p*-Value	Post hoc *p*-value
		No TBI-AC (*n* = 70)	TBI-AC (*n* = 50)		
Thrombomodulin (ng/mL)	11.0(8.6–12.6)	11.8(4.9–86.2)	10.2(3.9–84.0)	0.76	-
Syndecan-1 (ng/mL)	24.8(21.5–30.6)	29.9(19.2–39.5)	33.7(21.6–109.5)	0.03	HC vs. No TBI-AC 0.20 **HC vs. TBI-AC 0.04** **No TBI-AC vs. TBI-AC 0.03**

Continuous variables were reported as median (interquartile range), and compared using Wilcoxon rank-sum test. Bold depicts significant *p*-value.

**Table 3 medsci-06-00005-t003:** Glycocalyx shedding and endothelial disruption.

Clinical Parameters	No TBI-AC (*n* = 70)		*p*-Value	TBI-AC (*n* = 50)		*p*-Value
	No-Endotheliopathy (*n* = 38)	Endotheliopathy (*n* = 35)		No-Endotheliopathy (*n* = 21)	Endotheliopathy (*n* = 26)	
Thrombomodulin (ng/mL)	11.6(5.6–86.6)	10.8(3.9–21.7)	0.65	5.5(1.6–10.6)	20.2(8.0–88.0)	**0.002**

Continuous variables are reported as a median (interquartile range), and compared using the Wilcoxon rank-sum test. Bold depicts significant *p*-value.

**Table 4 medsci-06-00005-t004:** Comparisons of clinical outcome with respect to endotheliopathy in patients with/without traumatic brain injury-induced coagulopathy.

Clinical Parameters	No endotheliopathy (*n* = 61)		Endotheliopathy (*n* = 59)	*p*-Value
ICU length of stay (days)	6(3–11)		5(2–7)	0.26
Hospital length of stay (days)	8(4–17)		7(4–20)	0.49
PRBC (*n* = 57)	No	40(65.6)	21(35.6)	**0.001**
	Yes	21(34.4)	38(64.4)	
FFP (*n* = 34)	No	53(86.9)	33(55.9)	**<0.0001**
	Yes	8(13.1)	26(44.1)	
PRC (*n* = 25)	No	54(88.5)	41(69.5)	**0.01**
	Yes	7(11.5)	18(30.5)	
Sepsis	No	52(85.3)	49(83.1)	0.74
	Yes	9(14.8)	10(16.9)	
48-h mortality	No	57(93.4)	47(79.6)	**0.02**
	Yes	4(6.6)	12(20.4)	
30-day mortality	No	51(83.6)	33(55.9)	**0.001**
	Yes	10(16.4)	26(44.1)	

Categorical variables were reported as frequency (percentage) and the differences were tested by χ^2^ or Fisher’s exact test. Bold depicts significant *p*-value.

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
