# Peer review of "Acute Traumatic Endotheliopathy in Isolated Severe Brain Injury and Its Impact on Clinical Outcome"

_medsci, 2018, doi:10.3390/medsci6010005_

Reviewer 1 Report

Thank you for the opportunity to review this manuscript. Please see my comments below, which I believe would strengthen the manuscript:

General composition

There are some sentences and phrases in bold, and I’m not sure why (e.g. lines 47-49; line 70, line 136). Text within paragraphs should not be in bold

The text requires several grammatical corrections. For a couple examples, Line 63: I would change to “The endothelial glycocalyx (EGL), which was first postulated by Danielli in 1940, is 0.2……” Also Line 66: “serve as sensitive” should be “serves as a sensitive”. Line 86: “guardian’s” should be “guardians”. I won’t add anymore examples here, but would recommend that the whole manuscript is double checked for correct grammar.

Abstract

I think you should show the units for syndecan-1 (I assume it was ng/ml?)

Are the numbers is parentheses IQR or range? The authors should specify

Thrombomodulin is mentioned in the study design of the abstract but not the results.

Introduction

Line 49: bleed less than what?

The authors discuss the endothelial glycocalyx and syndecan-1 in some detail, but only a passing mention of thrombomodulin. It is well established that syndecan-1 is a good marker of glycocalyx shedding, but how did the authors chose thrombomodulin? Just a brief explanation would enhance the Introduction.

Methods

Line 92: the authors say ages 514 TBI patients aged 16-65 yrs, but then in Figure 1 it says that those aged<20 or="">50 were excluded. These do not match up. Also, why exclude >50 yrs? The authors need to justify this.

Line 92: “we screened 514 TBI”… but in Figure 1 this is 514 severe TBI. Did you screen all TBI, or just severe TBI?

Line 96: SABP needs to be written in full before being abbreviated

The authors have defined coagulopathy according to PT, INR, and aPTT. More modern trauma care would use ROTEM or TEG. Does the authors’ institution use this? If not, this should be noted in the manuscript so that the reader is more likely to accept the authors’ definition of coagulopathy as a pragmatic approach

The authors use thrombomodulin levels in their power calculation, but then go on to define endotheliopathy according to syndecan-1 levels, including in their ROC analysis. Can the authors justify their use of thrombomodulin in the power calculation, and then their switch to prominence of syndecan-1 throughout the text?

In the statistical methods section, the authors state that they will report categorical variables as frequency (%); however, they do not do this in the Results. For example in lines 151-152, the authors report % (N). Please make Results consistent with Methods.

Results

Lines 151-152: the average age of patients (35.2) is different to the abstract (35.7), and then AGAIN there is a different number in Table 1 (35.5)… are these data transcription errors? The abstract, results, and tables should be consistent! Also the % of male patients in Table 1 and Results (88.3%) is different to the abstract (96%). Such straightforward mistakes make this reviewers look at the remainder of data with some caution. During the revisions process, please double check all data.

In the clinical outcomes section of results (and Table 4), the authors discuss mortality, length of stay, transfusion requirements, sepsis, and then various permutations of mortality (48h, 30 days). This seems like mortality was the primary outcome measure, and the remainder were secondary outcome measures. These secondary outcome measures need to be listed in the Methods section to match up to the results (perhaps in the "Study variables" section, lines 131 – 136).      

Lines 255 – 266: The authors state that endothelial cell damage has not yet happened on admission. However, some investigators have shown that endothelial damage might happen in trauma patients within minutes of injury, before admission to hospital (Naumann et al, Shock doi: 10.1097/SHK.0000000000000999. [Epub ahead of print])

Table 1

This table requires several modifications:

The table requires another column of p-values that arise from the comparison between TBI-AC and No TBI-AC

GCS should not be reported as mean and standard deviation, because it is an ordinal scale, and should be treated as non-normal data, with median and interquartile range.

The authors are inconsistent with reporting N within parentheses, for example (n=10) or (n 10). Please use one or the other, but not both.

There are several items in Table 1 that are not mentioned in the Methods section. For example, length of stay. The methods and results ought to match up perfectly.

In line 167 the authors duplicate some data from Table 1 (haemoglobin and latelet levels), which is unnecessary.

Table 2

The authors use both HV and HC to indicate heathy volunteers. Please be consistent.

Discussion

In general, the discussion is a bit disjointed, and seems at times to be more of a summary of other studies that examined endotheliopathy. the authors need to summarise the data already in the literature, but relate it to their own data with greater contextual clarity and relevence.

Line 297: do the authors mean “association”, rather than “correlation”?

Line 300: “up to 400” is incorrect when there were 404 patients in the cited study

When discussing different cut-off values for syndecan-1 in severe TBI (lines 316 – 320), the authors should try to account for these differences. Why are the current study levels lower, and less different to heathy controls than in other studies of non-brain trauma? Could the mechanisms for glycocalyx shedding be different in brain injury than for haemorrhagic shock from a liver laceration, for example?

Lines 343-344: “Syndecan-1 is a potential biomarker of endothelial damage and was found to play a role in the development of early coagulopathy following brain trauma”. The authors cannot state this with the data presented. They have not proven any role of endothelial damage in the development of coagulopathy. They have only found associations, which can certainly generate the hypothesis of a mechanistic link, but cannot prove one.

Author Response

Dear Sir,

Kindly find all the answers and changes made mentioned in the attached pdf file.

Thanking you

with regards

arulselvi

Reviewer 2 Report

Dear editors, thank you for inviting me to review this intriguing study!

In my opinion, the study deals with a clinical and theoretical problem of great importance, it is generally well performed, it is based on a large patient material and the manuscript is well written. The conclusions drawn from this study are extremely interesting, e.g. almost twice as high syndecan-1 levels in non-survivors compared to survivors, and the significantly increased requirements for blood transfusion in patients with TBI endotheliopathy. However, I have a few remarks and questions to the authors.

1.     Page2, line 57: In my opinion “ameliorating thrombin generation” is not an optimal term to describe this process (“boost” or “rise” may be better).

2.     Page 3, line 93-94. Exclusion criteria – patients receiving blood or fluid prior to blood sampling was excluded. How many patients were excluded because of fluid infusion? In our practice, almost all the patients arriving to ER have fluid infusion started during the ambulance transport – that means we would not be able to include any patient. In my opinion, small amounts of iv. fluid would not affect the results.

3.     Page 3, line 106-107 + Fig.1. According to your definition of isolated TBI, the patients did not have any extracranial injuries. In our experience, the majority of severe TBI cases also have minor extracranial injuries. For instance, AIS score (extracranial injuries)<3 is often used to define isolated TBI.

4.     Page 4, line 123-125. Storage of samples at -20 C – is this temperature low enough to preserve the analyzed substances syndecan-1 and soluble thrombomodulin? Did you check the stability of samples during the storage?

5.     Page 6, Table 2. Please include more detailed legend to this and other figures/tables. For instance, does “HV” mean “healthy volunteers”? Please provide explicit explanations to the tables/figures.

6.     Page 9, line 250-256. This section needs to be re-written to improve its clarity. Generally, the relationship between syndecan-1 levels and thrombomodulin levels seems to be rather complicated, and it follows opposite pattern in patients with/without TBI-AC. I can understand the difficulty to describe this relationship and the possible pathophysiologic mechanisms behind it, but I wish to see some revision in the discussion section to make it more “didactic”.

My compliments to the authors for the excellent performed study, with recommendation for a minor revision!

Author Response

Dear Sir

Kindly find the reviewers comments with reply attached.

with regards

arulselvi

Round  2

Reviewer 1 Report

Thank you for addressing my comments in the revised manuscript. I still have some issues with the manuscript:

Tables

In the clinical parameters column of Table 1, you do not need to state the "normal range". Only the units are required. For example hemoglobin, red blood cell count, haematocrit, platelet count, white blood cell count

Please indicate in Table 2 that the P-value is for comparison of the two clinical groups (no TBI-AC vs. TBI-AC).

In the Post-hoc column of Table 2, either make the title of the column "post-hoc p-values", or within the data cells write p=0.03 rather than just 0.03.  

In Table 2, isn't the third post-hoc analysis for syndecan-1 just the same as the initial comparison and p-value (no TBI-AC vs TBI-AC; p=0.03)? This is a duplication, and can be deleted.

In Table 4, there are some text bold and some not. Was this intentional?

Results

The authors need to specifically state somewhere in the Methods or Results that they chose to perform ROC curve analysis for syndecan-1 because there was no significance with thrombomodulin. This will make it a nicer narrative, with fluency between steps in analysis.

I don't understand why the authors sometimes say NS for non-significant, and then at other times state the non-significant p-value (such as p=0.32). I would strongly recommend listing all of the raw p-values, and do not use the term NS anywhere in the manuscript, since this seems inconsistent.

Language

There are considerable gramatical issues throughout the text, which should be checked by a native English speaker / copy-editor before publication. They are not listed here, since I have concentrated on the scientific aspects of the manuscript only.

Author Response

Dear Sir,

Kindly find the reply to the comments / corrections mentioned.

Thanking you

arulselvi
